# Investigating the Spatial Heterogeneity and Influencing Factors of Urban Multi-Dimensional Network Using Multi-Source Big Data in Hangzhou Metropolitan Circle, Eastern China

Jing Zhang [1], Lei Li [1], Congmou Zhu [2,*], Qi Hao [1], Xinming Chen [3], Zhoulu Yu [4], Muye Gan [1] and Wuyan Li [5]

[1] Institute of Applied Remote Sensing and Information Technology, College of Environmental and Resource Sciences, Zhejiang University, Hangzhou 310058, China; zj1016@zju.edu.cn (J.Z.); 22114167@zju.edu.cn (L.L.); 22014175@zju.edu.cn (Q.H.); ganmuye@zju.edu.cn (M.G.)

[2] Collaborative Innovation Center of Statistical Data Engineering Technology & Application, Zhejiang Gongshang University, Hangzhou 310018, China

[3] Territorial Consolidation Center in Zhejiang Province, Department of Natural Resources of Zhejiang Province, Hangzhou 310007, China; 0922664@zju.edu.cn

[4] Zhejiang Digital Governance Space Planning and Design Co., Ltd., Hangzhou 310000, China; yuzl@zju.edu.cn

[5] The Institute of Land and Urban-Rural Development, Zhejiang University of Finance and Economics, Hangzhou 310018, China; liwuyan@zufe.edu.cn

[*] Correspondence: congmouzhu@zjgsu.edu.cn

**Abstract:** Exploring the spatial heterogeneity of urban multi-dimensional networks and influencing factors are of great significance for the integrated development of metropolitan circle. This study took Hangzhou metropolitan circle as an example, using multi-source geospatial big data to obtain urban population, transportation, goods, capital, and information flow information among sub-cities. Then, spatial visualization analysis, social network analysis, and geographical detector were applied to analyze the differences in spatial structure of multiple urban networks and influencing factors in Hangzhou metropolitan circle, respectively. The results showed that (1) the network connections of population, traffic, goods, and capital flows transcended geographical proximity except that of information flow, and population and traffic flow networks were found to be more flattened in Hangzhou metropolitan circle than in other urban networks; (2) the comprehensive urban network of Hangzhou metropolitan circle was imbalanced across sub-cities, presenting hierarchical and unipolar characteristics; and (3) the influence of traffic distance on the network spatial structure of Hangzhou metropolitan was stronger than the geographical distance, and the interactions between traffic distance and socioeconomic factors would further enhance the regional differentiation of the network spatial structure. This study could provide scientific reference for constructing a coordinated and integrated development pattern in a metropolitan circle.

**Keywords:** urban network; spatial structure; influencing factor; multi-source big data; Hangzhou metropolitan circle

## 1. Introduction

A metropolitan circle is a regional network with a megacity as the core where cities are spatially and functionally connected [1,2]. It has become an important spatial development strategy for competition in the global economy to reconstruct the metropolitan area across cities or provincial boundaries through the construction of intercity infrastructure and regional cooperation [3]. Its purpose is not only to expand the scale of population and urban space but also to increase the mobility of population, goods, capital, and information to strengthen intercity connections and improve resource allocation efficiency [2]. The spatial structure of a city or a region refers to the arrangement of a certain space in the urban form composed of population, materials, information, and a series of relationships generated by their interactions [4,5]. With the rapid progress of information technology and the

internet, the complex interaction between real space and virtual networks has enormously reshaped the metropolitan circles' spatial structure [6,7]. Castells [8] put forward the key concept of the "flow space", indicating that regional development increasingly depends on network interactions of various element flows. China had long adopted a strategy of giving priority to the development of urban agglomerations [6], but the construction of metropolitan circles had been neglected. This phenomenon has led to a series of development constraints in metropolitan circles, such as insufficient links between cities, the weak influence of core cities, and obstacles to the implementation of urban agglomeration planning [2,9]. According to *Outline of the 14th Five-Year Plan (2021–2025) for National Economic and Social Development and Vision 2035 of the People's Republic of China* (the 14th Five-Year Plan), China was striving to build a system of metropolitan areas centered on provincial capitals or central cities in order to improve the quality of urbanization and promote regional integrated development during the 14th Five-Year Plan period. Thus, it is of great significance to decipher the heterogeneity of spatial structure and influencing factors of metropolitan circles from the perspective of various "flow space", which are critical for optimizing the spatial structure of metropolitan circle and promoting regional integration and coordinated development.

The traditional view holds that the spatial structure of a city or region reflects regional differences and similarities in socioeconomic, political, or cultural attributes [10,11]. Early studies on regional spatial structure aimed at distinguishing clustering areas with similar attributes (e.g., natural and economic status) by using factor analysis and cluster analysis [12]. It allows us to reveal regional differences and identify specific morphologies of regional spatial structure, such as single-core, sectorial, or multi-core configurations [13]. The second view concentrates more on the spatial agglomeration and dispersion of regional structure, cartography, and spatial analysis, such as spatial auto-correlation analysis, geographically weighted modeling, and hotspot detection, are usually used [14,15]. Compared with the traditional views, this viewpoint simultaneously focuses on the attributes of specific spatial units and their spatial dependencies with adjacent units, thereby identifying the core and edge structure of an area [16]. However, it tends to measures local connections, but ignores the structural configuration of the entire area, without focusing on the spatial association between the target unit and nonadjacent unit [17]. In the era of globalization and information, regional interactive relationships have transformed from focusing on neighboring space to flow space. From the perspective of "flow space", the spatial structure of a city or region is viewed as an interconnected network, and more attention is paid to the nodes, backbone, and connections of the network [7,18]. It provides a new understanding of regional spatial structure, that is, city or sub-city divisions are nodes, and various element flows are links [19,20]. Based on the network perspective, the spatial relationships between two units are determined by a network composed of multiple element flows, rather than predetermined by the similarity of attributes or proximity of geographic distances [21,22].

There are various networks in regional spatial development, including natural connections, economic connections, cultural links, and administrative links [23,24], which form a complex connected network. Recent network-based studies concentrate on the spatial structure of a single network in a city or a region, such as the population flow network [25], goods flow network [26], information flow network [27], and traffic flow network [28]. These empirical studies have explored the specific roles of particular nodes [11], the spatial arrangement of various links, and the network organization of node city on the urban agglomeration or national scale [29–31]. A series of methods including network visualization and mapping technology, gravity models, and social network analysis have been developed [22,23,32]. However, existing related studies pay less attention to the scale of the metropolitan circle, and neglect the spatial heterogeneity of urban multi-dimensional flow networks [27,31]. Regional spatial structure is determined by various flow networks [33]. By only relying on the measurement of a single flow network, we cannot accurately position the role and competitiveness of a city in a regional network. Moreover, it is not conducive for providing effective decision support for formulating spatial structure optimization and

development strategies in the metropolitan circle. Many studies apply prefecture-level city network data to explore the spatial structure of a large region, such as the intercity transportation infrastructure, relational data for company headquarters and branches, intercity linkages for population, and goods mobility [34]. However, it is not adequately accurate to investigate the large-scale regional network structure using prefecture-level cities as spatial units, which may ignore the variations in the connections among sub-cities. Especially in China, the administrative scope of a city often includes several districts and counties. In the era of information, multi-source big data with high-resolution locational information are emerging, such as social media data, mobile phone signal data, and POI data [6,35,36]. These big data provide various flow network information among sub-cities that can contribute to the analysis of county-scale spatial structure.

The spatial structure of urban multi-dimensional networks in metropolitan circles is comprehensively influenced by the social economy and proximity factors [37]. In socioeconomic terms, the development level of economic, urbanization rate and population size are the core factors that affect the interactions between sub-cities [25,38]. Industrial development and financial investment affect financial and economic exchanges and cooperation between cities [39], further influencing the city's influence and service level in the surrounding areas. Meanwhile, the level of social consumption is an important intermediary and complementary factor for the interaction between cities [31]. These socioeconomic factors significantly influence the role of a city in regional network structure to a certain extent. The first law of geography states that all things are related, and things in close proximity are more closely related to each other, so that cities that are close to each other are more frequently connected [40,41]. Furthermore, the improvement in transportation has shortened the "space–time" distance between cities, and the proximity of industries also may strengthen the network links between cities, which can be considered the "industrial" distance [39]. However, to our knowledge, existing studies have usually focused on socioeconomic factors [39,42], and the influence of multi-dimensional distance on regional spatial network connections is easily ignored [43]. This is a clear and critical research gap, especially as the rapid progress of information technique and transportation in China promotes closer network connections between cities. Investigating the determining factors of the spatial network structure of various flows could provide policy supports for the integrated development of metropolitan circles.

To address these research gaps, this study attempts to consider the Hangzhou metropolitan circle, an area with early formation and rapid development, as an example, and then uses multiple internet data application platforms and web crawler technology to obtain five kinds of representative flow network information in Hangzhou metropolitan circle, including population, traffic, goods, information, and capital flows. Spatial mapping analysis, social network analysis, and geographical detector are applied to systematically explore the spatial networks of the metropolitan circle and the determining factors. The remaining part of this article is organized as follows. The second part presents the study area and data sources, the third part describes the various research methods adopted in this study, the fourth part describes the major findings, the fifth part shows the discussion and policy implications, and the sixth part concludes the paper.

## 2. Materials and Methods

### 2.1. Study Area

The Hangzhou metropolitan area is composed of four prefecture-level cities that are Hangzhou, Shaoxing, Jiaxing, and Huzhou (Figure 1), which is located in the south of the Yangtze River Delta urban agglomeration and in the northern part of Zhejiang Province. The research objective is to be attained at the county level, including four urban districts and 20 sub-cities in four prefecture-level cities (Figure 1). The district is the most important economic and political center of a prefecture-level city, so it is regarded as the central city of a prefecture-level city, and the others are regarded as sub-cities. The division of Hangzhou's central city is based on the Hangzhou 14th Five-Year Plan. Hangzhou

metropolitan circle covers an area of 34,585 km², in the proportion of 33.21% of the total area of Zhejiang Province; it has a resident population of 23.28 million, which is 39.81% of the total population of Zhejiang Province; its GDP reached CNY 3.07 trillion in 2020, which is 47.5% of the total provincial GDP; and its per capita GDP was CNY 129,000, which was 1.2 times the provincial average. In 2007, the first joint meeting of mayors of the Hangzhou Metropolitan Economic Circle was held, marking the start of the construction of the Hangzhou Metropolitan Circle. In 2014, the National Development and Reform Commission (NDRC) approved a comprehensive reform pilot project for the economic transformation and upgradation of the Hangzhou Metropolitan Circle. Owing to long-term regional cooperation and coordinated development, the co-urbanization effect of these four cities has become increasingly significant. Hangzhou metropolitan circle has become one of the most populous and open regions in China. Spatial network elements such as population flow, goods flow, and information flow are closely related in this metropolitan circle. However, it also suffers from some development constraints, such as widening regional development gaps, weak regional division and cooperation, and a lack of regional coordination and consultation mechanisms [44]. Therefore, investigating the spatial heterogeneity and influencing factors of urban multi-dimensional networks in Hangzhou metropolitan circle is of strategic significance to promote the integration and coordinated development among sub-cities and form an orderly cooperative relationship within the region.

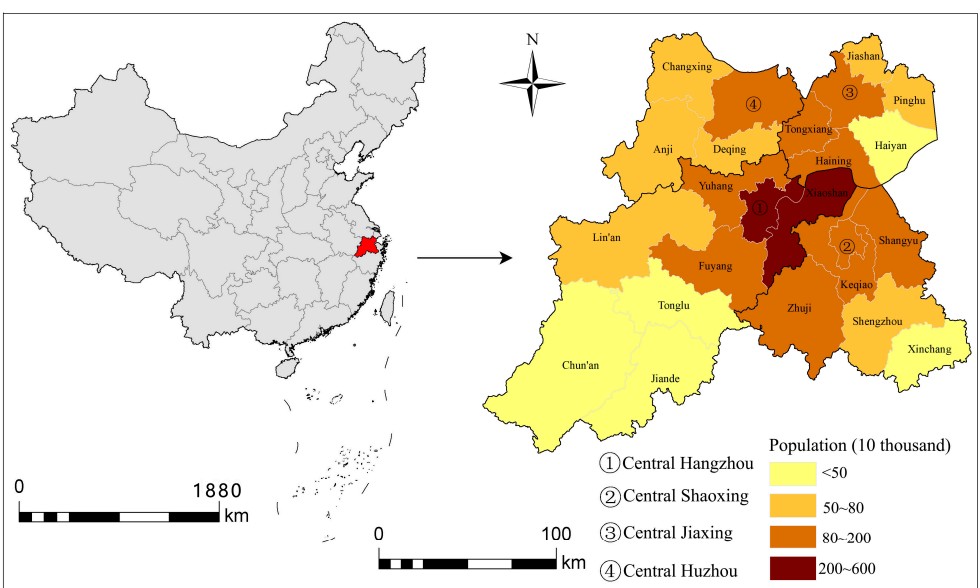

**Figure 1.** Location of four prefecture-level cities and 24 sub-city divisions in the Hangzhou metropolitan circle, China.

*2.2. Data Sources and Preprocessing*

In this study, the following five urban big data were used to identify the spatial networks of the metropolitan circle.

(1) Human flow: Population flow is the most frequent connection between sub-cities. Communication is the demand of human daily behavior, and Baidu's migration platform could map the population flow trajectory and spatial distribution through the positioning information of Internet users based on the positioning service technology. The population flow data in this platform has been widely used in China's population flow research [45]. In our study, the population flow data were collected from 1 October 2020 to 18 January 2021. Considering that the data reflects only the relative value of in-migration and out-migration, this paper used the travel index to calculate the floating population matrix among four prefecture cities in Hangzhou metropoli-

tan circle. Then, the population proportions within the 24 sub-cities were counted to construct a 24 × 24 population proportion matrix. Finally, the relative mobile population value between sub-cities was calculated to construct a 24 × 24 human flow matrix [46].

(2) Goods flow: Goods flow refers to the process of transferring goods from the supply location to the receiving location. In this study, the POI points of all logistics expressed in the region were used to construct the logistics matrix. The data were collected in 2020 from 10 major logistics companies in the study area, including ShunFeng, Zhongtong, Yunda, Jingdong, Tiantian, Shentong, Yuantong, Deppon, Baishi, and Jutu. In this study, the method proposed by Taylor [47] was used to construct a 24 × 24 goods flow matrix.

(3) Capital flow: Similar to the goods flow data, all branch institutions of five banking companies in the Hangzhou metropolitan circle, including the Industrial and Commercial Bank of China, Bank of China, Agricultural Bank, Construction Bank, and Postal Savings Bank, were selected for this study [33]. Based on the chain model in urban network analysis, bank POI points were divided into five categories and assigned different values to construct a 24 × 24 capital flow matrix [48].

(4) Information flow: With the development of internet technology, information connection has become an increasingly important method of communication between sub-cities, and sub-cities within metropolitan circles are organically connected through networks.58.com (accessed on 1 September 2023) website is the largest classified information website in China, and has the unique advantage of mutual search among sub-cities [46]. Therefore, this study used the average daily search volume of 58.com in 2020 by using the selenium library in Python to construct a 24 × 24 information flow matrix.

(5) Traffic flow: Traffic flow reflects the connection intensity of sub-cities in the transportation network. The railway is the most important and popular transportation mode for Chinese residents. In this study, the information on the number of station locations and passing trains was obtained from the China Railway Customer Service Center, which is the largest railway information publishing website in China. The distance traveled by car between sub-cities was obtained from Gaode Map. These two kinds of data were jointly applied to construct a 24 × 24 traffic flow matrix [49].

*2.3. Methods*

2.3.1. Strength Analysis of Multiple Urban Flow Networks

Based on the above matrix information of multiple flow networks between sub-cities, the flow space network measuring method proposed by Taylor et al. (2004) was applied to measure the connection strengths of various urban flow networks [47]. Furthermore, we normalized the flow connection strength and summed these data to measure the strength of the comprehensive flow network of the metropolitan circle. The specific formula was as follows:

$$
\begin{aligned}
S &= R_{ij} \times R_{ji} \\
S' &= (S - S_{\min})/(S - S_{\max}) \\
F_i &= \sum_j S'_{ij}
\end{aligned}
\tag{1}
$$

where $S$ denotes the connection strength of the flow network between sub-city $i$ and $j$, $R_{ij}$ denotes the amount of "flow data" from sub-city $i$ to sub-city $j$, and $R_{ji}$ denotes the amount of "flow data" from sub-city $j$ to sub-city $i$. $S'$ is the normalized value of the network connection strength, $S_{min}$ is the minimum value of the network connection strength, and $S_{max}$ is the maximum value of the network connection strength. $F_i$ is the connection strength of the comprehensive flow network of sub-city i, and $S'_{ij}$ is the connection strength between sub-city $i$ and $j$. Considering that various urban flow networks are equally important

for the integration development of metropolitan circles, each element flow [31] is given a weight of 0.2.

2.3.2. Social Network Structure Analysis

(1)  Dominant flow analysis

In this study, the dominant flow analysis method was employed to identify the general spatial structural patterns of flow networks according to the maximal flow rank relationship [50]. The higher value indicates the leading position of a sub-city in the regional flow network. This study selected the top two orders (first and second) to identify the regional main backbone network. The specific formula was as follows:

$$PC = RC_i / \sum_n RC_i \tag{2}$$

where $PC$ is the relative centrality index, and $RC_i$ is the strength of the flow connection under the i-th order.

(2)  Connectivity influence degree

This method comprehensively measures the total influence of a sub-city in regional spatial structure. The higher the value is, the stronger the node effect of a sub-city, and the wider the network interaction range. This index has been widely used to reflect the influence of a sub-city in regional spatial networks [51]. The specific formula was as follows:

$$PI = \frac{\sum_{i=1}^{5} RC_i}{V_{AD} + 3V_{SD}} \tag{3}$$

where $PI$ is the influence index of a sub-city in regional flow networks, and $V_{AD}$ and $V_{SD}$ are the average value and standard deviation value of $RC_i$, respectively, and $RC_i$ is the flow connection strength between sub-cities under the i-th order.

(3)  Dominant structural index

Based on the spatial structure algorithm of regional poly-centricity proposed by Hanssens et al. [51], this paper employed a spatial structure dominance index to reflect the network dominance level. The index is in the range of 0 to 1, where 0 suggests a distinct unipolar development trend in the regional spatial structure, and 1 suggests obvious multiple polar features in regional spatial structure. The specific formula was as follows:

$$SSI = \begin{cases} (2 - SD/SD_{rc})/2, SD \leq SD_{rc} \\ SD_{rc}/2SD, SD > SD_{rc} \end{cases} \tag{4}$$

where SSI is the regional spatial structure index, SD is the standard deviation value of the connection strength of flow in a sub-city, and $SD_{rc}$ is the standard deviation value of the connection strength of flow in all sub-cities after ranking.

(4)  Visual analysis of combined geographic–topological networks

In this study, we visualized the flow connection by using the chord diagram, the length of the arc was used to represent the size of the element attribute, and the thickness of the connections among the nodes was used to reflect the strength of the connections, which can directly reveal the topological interactions among unit nodes. Meanwhile, the distribution patterns of multiple flow strength were superimposed with geographic space, and the structure of flow connection was displayed to explore the spatial networks' mechanism of the metropolitan circle by using the ArcGIS spatial analysis platform.

*2.4. Geographical Detector*

(1)  Model construction

In this study, we applied the geographical detector to identify the factors that influence the spatial structure of urban multi-dimensional networks in the metropolitan circle. The principle of geographic detectors is to assume that the independent variable and the dependent variable are similar in spatial distribution, and then the independent variable has a significant impact on the dependent variable [52,53]. This study used the factor detector and interactive detector modules to explore the formation mechanism of the urban multi-dimensional networks in the metropolitan circle. The detailed formula was as follows:

$$q = 1 - \frac{1}{N\sigma^2} \sum_{n=1}^{L} N_h \sigma_h^2 \tag{5}$$

where $q$ measures the interpretative power of the independent variable, $h$ is the stratification of the dependent or independent variable, and $N_h$ and $\sigma_h$ denote the number of sub-cities and variance in strata h, respectively. The $q$ value is in the range of 0 to 1. A higher q value represents a stronger explanatory power of the independent variable on the dependent variable.

Furthermore, the interactive detector was used to measure the interactions between driving factors and to reveal the impact of the joint effects of explanatory variables on the explanatory power of dependent variable. The judgment types of interaction between two factors can be classified into univariate nonlinear weakening, nonlinear weakening, bivariate enhancement, and independent and nonlinear enhancement [54].

(2)    Selection of driving factors

Based on the above analysis, this study selected twelve representative factors as drivers along four dimensions: population, economy, society, and distance. Population size (x1), size of working-aged population (x2), and population urbanization rate (x3) were selected to reflect the demographics of a sub-city; GDP per capita (x4), gross output value of enterprise above the designated size (x5), and total retail sales of social consumer goods (x6) were selected to measure the economic development of a sub-city; number of hospital beds (x7), road density (x8), and social security expenditure and employment fiscal expenditure (x9) were selected to measure the social development of a sub-city; and geographic distance (x10), traffic distance (x11), and industrial distance (x12) were selected to measure the distances between sub-cities. Specifically, geographical distance was measured by the linear distance between the administrative centers of sub-cities; traffic distance was calculated by the shortest railway transportation time; industry distance was represented by industrial structural similarity index [39,43].

## 3. Results

### 3.1. The Strength of Multiple Urban Flow Networks in the Metropolitan Circle

Figure 2 shows the connection strengths of multiple flow networks of 24 sub-cities within four (prefecture-level) cities in Hangzhou metropolitan circle. As shown, the connection strengths of the population and goods flow networks were generally higher than the others, and the connection strength of information flow network was the lowest. In terms of the five flow networks, the spatial connections of population, traffic, goods, and capital flows showed a network radial distribution pattern with Central Hangzhou as the core, indicating that the four flow networks transcended geographical proximity. The information flow connections between sub-cities were mainly distributed within the city, indicating that the information flow did not form an open network spatial structure within Hangzhou metropolitan area. In terms of the connection strength, the top three sub-cities with the strongest connection strength of population, information, and capital flow were the Central Hangzhou, Xiaoshan, and Yuhang, indicating that these sub-cities were the main concentrated areas of population, information, and capital in Hangzhou metropolitan area. Regarding the traffic flow, the connection strength in Central Shaoxing was stronger, it was attributed to the fact that Shaoxing city is the central hub of the Hangzhou and Ningbo metropolitan areas in Eastern China, and has complete high-speed and railway

transportation. In terms of the goods flow, Zhuji, one of the important production bases of small commodities in Zhejiang Province, became one of the top three sub-cities with the strongest connection strength, indicating that the rapid growth of online retail industry has greatly enhanced the position of Zhuji in the goods network.

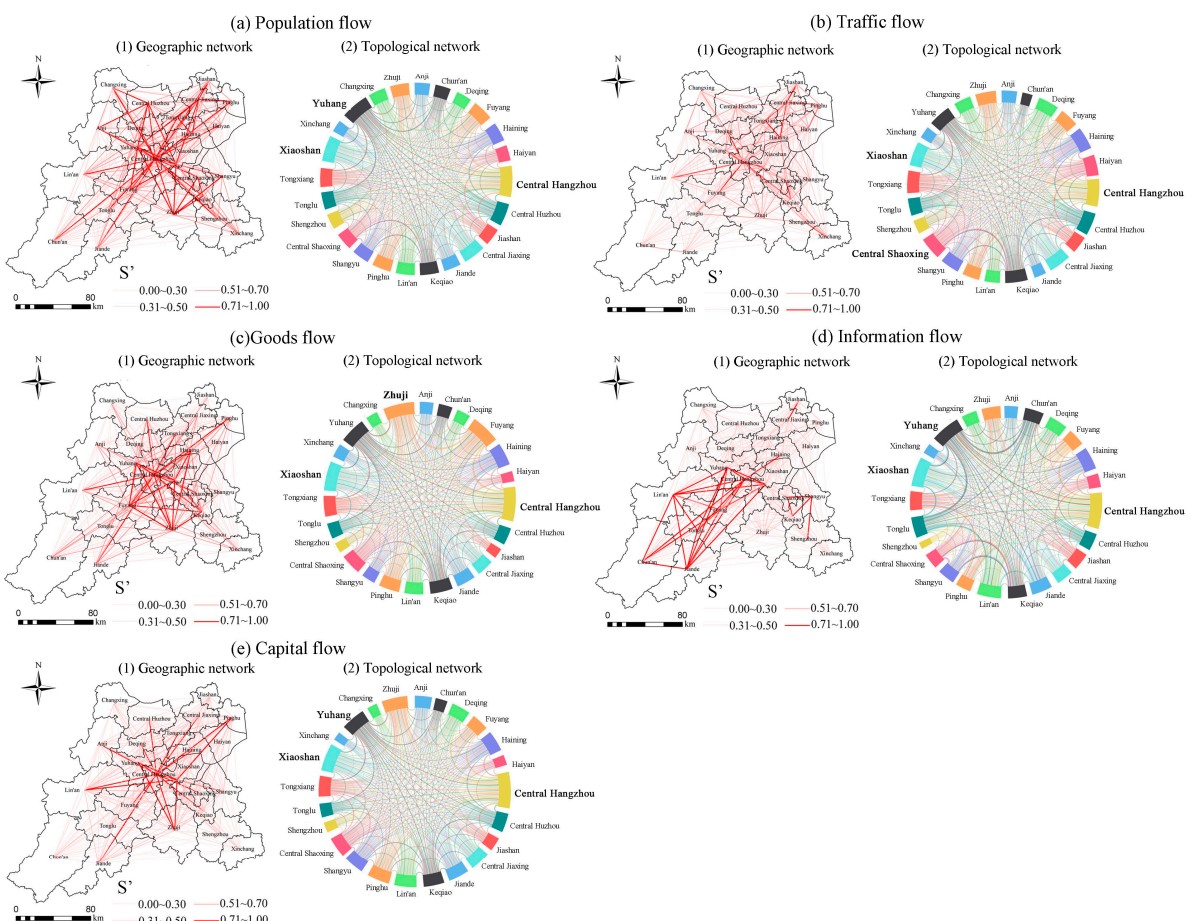

**Figure 2.** Strength of multiple spatial networks of Hangzhou metropolitan circle.

### 3.2. The Hierarchical Structure of Multiple Urban Flow Networks in the Metropolitan Circle

We used the connectivity influence degree to measure the connection influence of flow networks among sub-cities (Figure 3a–e). The high-value areas of connection influence of population and information flow networks were in Central Hangzhou, Xiaoshan, and Yuhang, indicating that the three sub-cities were dominant in the population and information flow networks of the Hangzhou metropolitan circle. Remote areas, such as Chun'an, Jiande, Xinchang, and Anji, showed low connectivity influence degree of regional population and information flow networks. The connection influence of traffic flow network in Central Hangzhou and Xiaoshan was at a high level, indicating that these two sub-cities took the lead in regional traffic flow network. While the low-value areas of traffic flow network were mainly distributed in the northwestern and southwestern areas. In terms of the goods network, the high-value areas of connection influence were in Central Hangzhou, Fuyang, and Zhuji, indicating that these three sub-cities had a greater influence on regional goods flow network. Regarding the capital network, Central Hangzhou, Xiaoshan, and Zhuji were the high-value areas of connection influence, indicating that the three sub-cities were in a leading position in regional capital network. While the low-value areas were located in the periphery of the metropolitan circle. Furthermore, the dominant structural index was used to identify the spatial hierarchical structure of multiple flow networks. The SSI value of the goods network was the smallest, at 0.295, followed by the capital flow network, indicating that the two flow networks presented a more pronounced unipolar

development trend. The SSI value of traffic flow networks was higher than 0.50, indicating that its structure was relatively discrete.

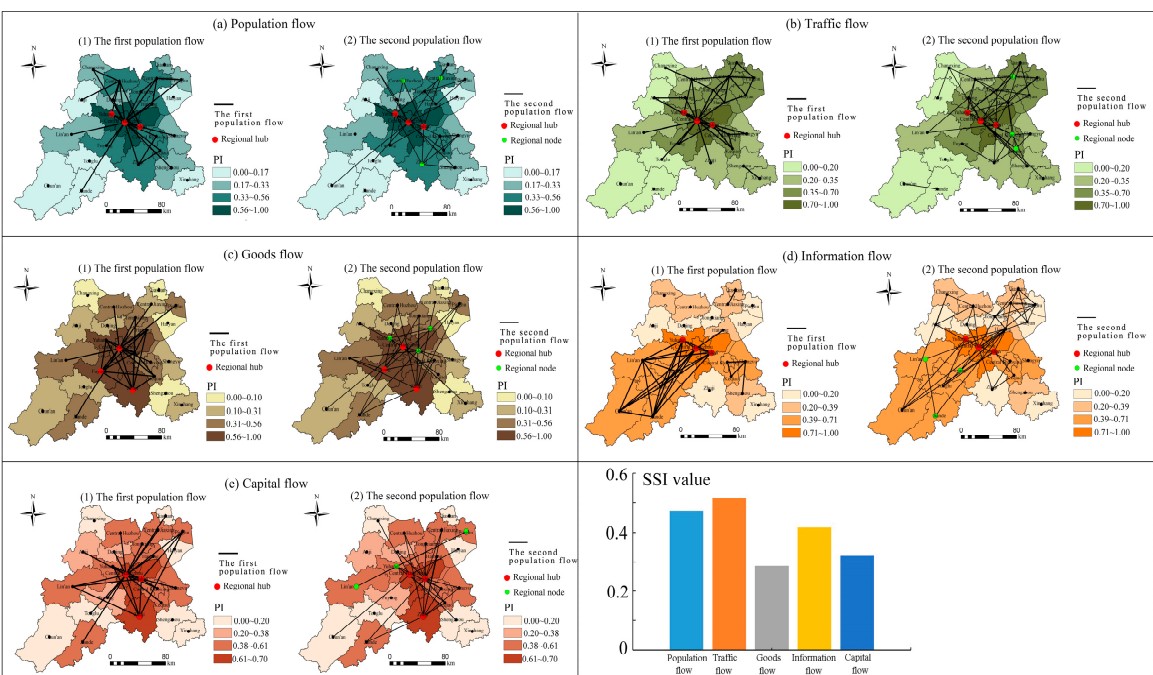

**Figure 3.** The connection influence and structure of multiple urban networks in Hangzhou metropolitan circle.

To further identify the structural model of multiple flow networks, the dominant flow analysis method was used to identify the first- and second-largest dominant flow networks (Figure 3). Specifically, the first-largest dominant networks of population, traffic, goods, and capital flow presented a radial network axis with Central Hangzhou, Xiaoshan, Yuhang, Zhuji, and Fuyang as the hubs. While the first-largest network of information flow was mainly concentrated in Hangzhou city, and considered Central Hangzhou, Xiaoshan, and Yuhang as the hubs, confirming that the influence of information flow has not yet overcome the restrictions of administrative boundaries. In terms of the second-largest dominant network, different element flows presented distinct patterns of spatial structure. The second-largest dominant networks of population and traffic flow were mainly distributed in the east and north of Hangzhou metropolitan circle, considering some sub-cities of Huzhou and Jiaxing as nodes. The second-largest dominant network of goods and information flow connected most sub-cities in a decentralized form, taking some sub-cities of Hangzhou as nodes. The second-largest dominant network of capital flow mainly distributed in Hangzhou and Shaoxing cities, and took Lin'an, Yuhang, and Pinghu as nodes.

### 3.3. Spatial Pattern of the Comprehensive Urban Flow Network in the Metropolitan Circle

The strength and pattern of the urban comprehensive flow network in Hangzhou metropolitan circle is described in Figure 4. The distribution of connectivity influence was imbalanced across sub-cities, presenting a hierarchical structure in the Hangzhou metropolitan circle. Three sub-cities, from Hangzhou, including Central Hangzhou, Yuhang, and Xiaoshan, had the highest values of connectivity influence on the comprehensive flow network, with the other sub-cities showing a decreasing trend in distance from Central Hangzhou. This indicated that the comprehensive flow network of the Hangzhou metropolitan circle presented an obvious spatial structure of core–periphery. These imbalanced distributions of comprehensive flow network were may be due to the varying locations, population, and economic sizes of sub-cities. In terms of the structural pattern of the comprehensive flow network, its first-largest dominant network considered Central

Hangzhou, Xiaoshan, and Yuhang as hubs, and was mainly distributed in Hangzhou and Shaoxing cities. This indicated that the connection between Hangzhou and Shaoxing was more close. The second-largest dominant network considered Fuyang, Keqiao, and Central Shaoxing as nodes, suggesting that the three sub-cities served as a "bridge" connecting the center and the peripheries.

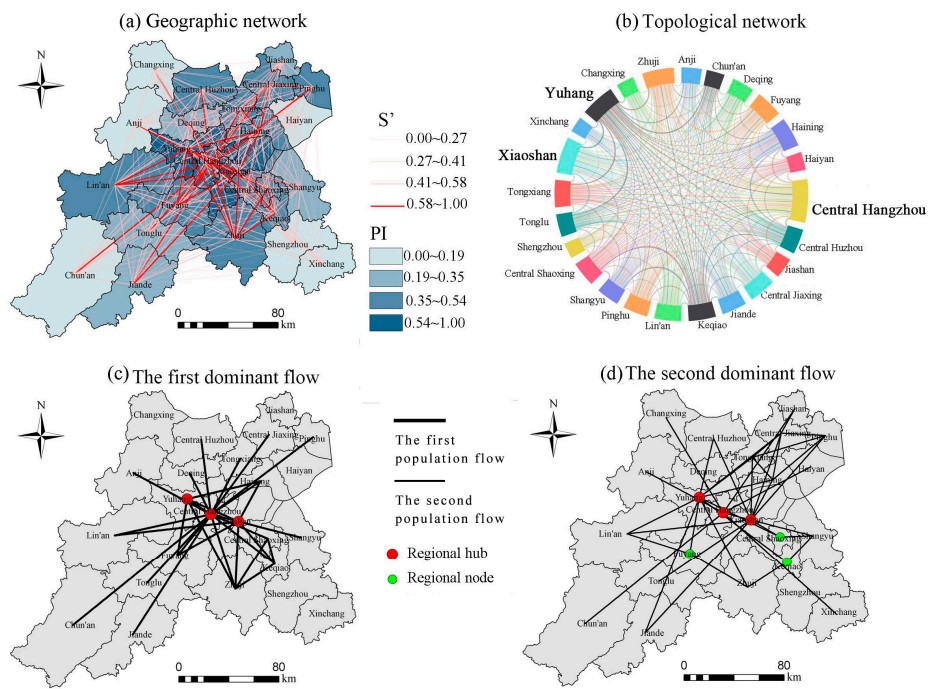

**Figure 4.** The strength and pattern of the comprehensive flow network in Hangzhou metropolitan circle.

### 3.4. Factors Influencing Comprehensive Flow Network of Metropolitan Circle

We used a geographic detector to identify the dominant factors on the distribution pattern of the comprehensive flow network in metropolitan circle. In this study, the dependent variable was the connection strength of the comprehensive flow network in Hangzhou metropolitan circle, and the independent variables included the twelve above-mentioned representative factors from four dimensions of population, economy, society, and distance. Significant influencing factors and their interpretative powers (q-values) are shown in Table 1.

**Table 1.** Influencing factor detection of spatial differentiation of the comprehensive spatial network.

| Variables | q Statistics | Rank | Variables | q Statistics | Rank |
|---|---|---|---|---|---|
| $X_1$ | 0.8398 *** | 2 | $X_8$ | 0.6562 ** | 7 |
| $X_3$ | 0.8353 *** | 3 | $X_9$ | 0.6094 ** | 8 |
| $X_5$ | 0.7302 *** | 6 | $X_{10}$ | 0.7561 *** | 5 |
| $X_6$ | 0.8535 *** | 1 | $X_{11}$ | 0.8275 *** | 4 |

Note: ***: $p < 1\%$, **: $p < 5\%$, where $p$ represents the level of significance, $p < 5\%$ means significant difference, and $p < 1\%$ means significant difference.

From a horizontal comparison point of view, the spatial differentiation of the comprehensive flow network was influenced by a combination of socioeconomic and geographic distance. The top four factors influencing the spatial differentiation of the comprehensive flow network were, in order, the total retail sales of social consumer goods, population urbanization rate, size of working-aged population, and traffic distance, which can be regarded as the dominant factors. This indicated that regional economic development, population growth, and transportation connections were the dominant factors affecting the connection strength of the comprehensive flow network of Hangzhou metropolitan circle.

An interactive detector was used to reveal the interpretative power of the interaction between two influencing factors on the spatial heterogeneity of the comprehensive flow network. Compared with the influence of a single factor, the interpretative power of most two-factor interactions was stronger (Table 2), indicating that the two-factor interaction enhanced the spatial heterogeneity of the comprehensive flow network of the Hangzhou metropolitan circle. Specifically, the interaction type of the population size and the other significant factors was bivariate enhanced, indicating that the interaction between population increase and other socioeconomic factors enhanced the spatial differentiation of the comprehensive flow network. The interaction between population size and gross output value of enterprise above the designated size was stronger than that with other factors. In the interaction between population urbanization rate and the other factors, their interaction type was bivariate enhanced, except for the interaction between population urbanization rate and road density. The interaction between population urbanization rate and total retail sales of social consumer goods was stronger, and the interaction between population urbanization rate and road density was the weakest. The main interaction type between the gross output value of enterprise above the designated size, the total retail sales of social consumer goods, and other factors was the bivariate enhancement type, indicating that the dual effect of economic development and other socioeconomic factors promoted the spatial differentiation of the comprehensive flow network. The interaction type between gross output value of enterprise above the designated size above the designated size and road density was single-factor nonlinear attenuation with the weakest status, indicating that the interaction between them weakened the regional differentiation of the comprehensive flow network. The interaction type of road density, social security and employment fiscal expenditure, and other factors was bivariate enhanced, and the interaction between road density and social security and employment fiscal expenditure was the weakest. The interaction type between geographic distance and traffic distance was also bivariate enhancement, indicating that the interaction between them enhanced the spatial differentiation of the comprehensive flow network.

**Table 2.** Interactive influence results of spatial differentiation of the comprehensive spatial network.

| $X_i \cap X_j$ | $q\,(X_i \cap X_j)$ | Interaction Type | $X_i \cap X_j$ | $q\,(X_i \cap X_j)$ | Interaction Type | $X_i \cap X_j$ | $q\,(X_i \cap X_j)$ | Interaction Type |
|---|---|---|---|---|---|---|---|---|
| $X_1 \cap X_3$ | 0.9125 | Bivariate enhancement | $X_3 \cap X_9$ | 0.8801 | Bivariate enhancement | $X_6 \cap X_9$ | 0.9013 | Bivariate enhancement |
| $X_1 \cap X_5$ | 0.9250 | Bivariate enhancement | $X_3 \cap X_{10}$ | 0.8925 | Bivariate enhancement | $X_6 \cap X_{10}$ | 0.9048 | Bivariate enhancement |
| $X_1 \cap X_6$ | 0.8621 | Bivariate enhancement | $X_3 \cap X_{11}$ | 0.8675 | Bivariate enhancement | $X_6 \cap X_{11}$ | 0.9301 | Bivariate enhancement |
| $X_1 \cap X_8$ | 0.8603 | Bivariate enhancement | $X_5 \cap X_6$ | 0.9626 | Bivariate enhancement | $X_8 \cap X_9$ | 0.7605 | Bivariate enhancement |
| $X_1 \cap X_9$ | 0.9018 | Bivariate enhancement | $X_5 \cap X_8$ | 0.7129 | Single-factor Nonlinear attenuation | $X_8 \cap X_{10}$ | 0.8742 | Bivariate enhancement |
| $X_1 \cap X_{10}$ | 0.9240 | Bivariate enhancement | $X_5 \cap X_9$ | 0.9581 | Bivariate enhancement | $X_8 \cap X_{11}$ | 0.8320 | Bivariate enhancement |
| $X_1 \cap X_{11}$ | 0.9152 | Bivariate enhancement | $X_5 \cap X_{10}$ | 0.8202 | Bivariate enhancement | $X_9 \cap X_{10}$ | 0.8250 | Bivariate enhancement |
| $X_3 \cap X_5$ | 0.9176 | Bivariate enhancement | $X_5 \cap X_{11}$ | 0.9266 | Bivariate enhancement | $X_9 \cap X_{11}$ | 0.9139 | Bivariate enhancement |
| $X_3 \cap X_6$ | 0.9208 | Bivariate enhancement | $X_6 \cap X_8$ | 0.8592 | Bivariate enhancement | $X_{10} \cap X_{11}$ | 0.8409 | Bivariate enhancement |
| $X_3 \cap X_8$ | 0.8187 | Single-factor nonlinear attenuation | | | | | | |

## 4. Discussion

Under the tide of globalization and information, the interactive relationships between cities have transformed from spatial proximity to a connection network dominated by time proximity [55]. Compared with the traditional statistical data-based spatial structure model, this study used the geospatial big data and social network analysis to identify the various urban flow networks of metropolitan circle, which can provide a new perspective and method to comprehensively evaluate the spatial development of the metropolitan

circle in the context of single flow-led studies [7]. Furthermore, considering that the connection strength between sub-cities is significantly affected by location conditions and socioeconomic development, this study used the geographical detector to identify the determining factors of the flow networks in the metropolitan circle, which contributes to providing references for the integration development of the metropolitan circle.

The results of our study indicated that the connection strengths of the population and traffic flow networks were higher, and this finding is confirmed in other related studies of metropolitan circles and urban agglomerations [25,26]. They pointed out that population and goods flows were the most frequent in the flow networks between cities. Furthermore, this study found that there were significant differences in the connection strength of different types of flow networks in Hangzhou metropolitan circle, especially the information flow, and its connection strength between sub-cities was limited by the administrative boundaries of prefecture-level cities. This phenomenon indicated that information network in Hangzhou metropolitan circle was not strong, reflecting the limited development of metropolitan circle [33]. Furthermore, the results showed that the flow connections between sub-cities from Hangzhou, Jiaxing, and Shaoxing was stronger than other sub-cities. This is because the local government proposed the development strategies of Hangzhou–Shaoxing and Hangzhou–Jiaxing integration development in 2018 and 2020, respectively. The construction of intercity transportation and public service facilities and the coordinated division of industries have strengthened the connection of multiple flows between these sub-cities [30,56].

Regarding the structural characteristics of various urban flow networks in Hangzhou metropolitan circle, the spatial urban network of the metropolitan circle presented hierarchical and unipolar characteristics. This characteristic is quite different from the dual-center or multi-center structure exhibited by urban agglomerations [39,55]. Corresponding with the Hangzhou metropolitan circle, three sub-cities from Hangzhou city, including central Hangzhou, Xiaoshan, and Yuhang, had the strongest connection influence in the flow networks, which means that various flow networks reinforce the existing urban system. This finding indicated that there is a positive correlation between flow network connections and the hierarchy of sub-cities with a metropolitan circle, where sub-cities with higher administrative levels would play stronger roles in urban network connections [57]. Additionally, this study also discovered that some sub-cities with advantageous industries played important roles as nodes in regional flow network. For example, Zhuji was regarded as the hub of goods network in Hangzhou metropolitan circle. This is due to the fact that there are many small commodity manufacturers in Zhuji, and the rapid development of the online retail industry has greatly enhanced its status in the goods network.

In the analysis of influencing factors, the dominant factors influencing the spatial differentiation of the network structure were population, social consumption, and traffic distance. This result is consistent with the findings of Zheng et al. [39]. Earlier work noted that in highly urbanized cities, social and economic activities are more prosperous, and various element flows tend to be more frequent [58]. With the continuous improvement of traffic infrastructure, the impact of traffic distance on the spatial differentiation of the regional flow network structure was found to be stronger than that of geographical distance, which is in line with Chen et al. [59]. Furthermore, the results also suggest that the interactions between socioeconomic and traffic distance have an influence on reinforcing the status of cities. This is because that in the process of regional development, the interactions between socioeconomic activities and transportation infrastructure can accelerate the flow and aggregation of various elements, promoting the status of a city in the regional network structure [39,60].

Based on the above study results, some policy implications to promote the integrated development of metropolitan areas could be suggested. First, using multi-source big data, this study found that the connection strengths of the population, goods, and traffic flow networks between sub-cities were strong, while the connections of the information and capital flow networks were relatively weak. Therefore, the government ought to increase

investment in information network infrastructure and apply information technologies such as artificial intelligence, cloud computing, and big data to accelerate the development of digital financial networks [49]. Second, according to the results of this study, the spatial structure of urban network in Hangzhou metropolitan circle was characterized by a core–periphery structure. Numerous population, capital, technology, and other resources flow into Central Hangzhou, resulting in limited development of marginal areas. Therefore, in order to transform the regional spatial structure into a balanced "network–node" pattern, the government should support the free-flow enhancement of the radiation and driving role of hubs and core nodes, and focus on developing the linkage ability of peripheral sub-cities to connect with extra-regional networks [33,51]. Finally, the results showed that traffic distance and geographical distance were found to have significant impacts on the spatial network structure of metropolitan circles, and the interactions of traffic distance with population size and economic development would reinforce their influence. Thus, local governments could promote the development of transportation infrastructure represented by high-speed railways, expressways, and intercity intelligent transportation, which can greatly shorten the space-time distance between sub-cities. Additionally, provincial government could innovatively develop more cross-regional and closely related group areas to rationally allocate population and industries, information infrastructure and public service facilities can be improved to accelerate the flow and connection of regional resources. These series of cross-regional construction measures contribute to promoting the polycentric development of regional spatial structure.

The contributions of this study to the analysis of regional flow networks include two aspects. First, this study employed multi-source big data to systematically investigate the connections and structures of various flow networks between sub-cities in the metropolitan circle, which is conducive to analyze the spatial structure of metropolitan circles and its differences from the perspective of different element flow. Most studies identify regional spatial structure by using data from a single flow category [22,55], and fail to reveal the multi-dimensional spatial network structure. The second contribution is that this study quantitatively explored the distance and socioeconomic factors that influence the spatial differentiation of the connection strength of comprehensive flow network, which could provide policy inspiration for promoting the integrated development of metropolitan circles. It is worth pointing out that as far as we know, there are few studies on the analysis of the determinants of regional flow network connections combining geographic, demographic, social, and economic dimensions as a result of the limitation of big data and methods. This study is an attempt to fulfil these research gaps in this regard.

Overall, this study also faces two limitations. First, although the big data used in this study are for the sake of data availability and convenience, the accuracy of the results is still slightly insufficient. For example, the population flow ignores the travel data of private cars, and the goods flow does not consider large-scale material transportation. Thus, future research needs to refine the comprehensiveness and accuracy of the data. Second, this study does not consider the temporal evolution of the spatial network structure of the metropolitan circle, and it is essential to reveal the internal mechanism of regional spatial structure from a time-across perspective [61]. Future research needs to investigate the spatiotemporal characteristics of the spatial network structure of metropolitan circles based on spatiotemporal panel data.

## 5. Conclusions

The connection strengths and spatial structure of multiple flow networks between sub-cities in metropolitan circles is a critical basis to evaluate the integrated development of metropolitan circles. Considering the Hangzhou metropolitan circle as a case study area, this study examined the spatial network structure of the metropolitan circle based on multiple big data, and determining factors influencing the spatial differentiation of the comprehensive flow network between sub-cities in metropolitan circle were identified using a geographical detector.

The following are the main findings of this study. The connection strengths and spatial distribution vary in different types of elements in the metropolitan circle. Central Hangzhou, Yuhang, and Xiaoshan from Hangzhou city as well as some sub-cities with characteristic industrial advantages were identified as regional nodes in different flow networks. The distribution of the connectivity influence on comprehensive flow network presented hierarchical and unipolar characteristics in the metropolitan circle, which is quite different from the multi-center structure of urban agglomerations. The spatial differentiation of the connection strength of comprehensive flow network was strongly affected by distance and socioeconomic factors in Hangzhou metropolitan circle, especially the traffic distance, and its interactions with socioeconomic factors would strongly enhance the spatial differentiation of flow network.

**Author Contributions:** Conceptualization, J.Z. and C.Z.; methodology, L.L. and Q.H.; software, Z.Y.; formal analysis, W.L.; investigation, X.C.; writing—original draft preparation, M.G. and C.Z. All authors have read and agreed to the published version of the manuscript.

**Funding:** This research was funded by the Basic scientific research business cost project of Zhejiang Gongshang University, grant number XT202209, the National Natural Science Foundation of China, grant number 41701171, and the Open Fund of Key Laboratory of Urban Land Resources Monitoring and Simulation, Ministry of Natural Resources, grant number KF-2020-05-073.

**Data Availability Statement:** Not applicable.

**Conflicts of Interest:** The authors declare no conflict of interest.

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
