# Peer review of "Investigating the Spatial Heterogeneity and Influencing Factors of Urban Multi-Dimensional Network Using Multi-Source Big Data in Hangzhou Metropolitan Circle, Eastern China"

_land, doi:10.3390/land12091808_

Round 1

Reviewer 1 Report

This manuscript presents a Study on “Investigating the spatial heterogeneity and influencing factors of urban multi-dimensional network using multi-source big data in Hangzhou metropolitan circle, eastern China.” There are, however, many inconsistencies within the manuscript that should be clarified before acceptance for publication can be recommended.

1. English should be checked. There are grammar and function errors.

2. What is the originality of this study?

3. The abstract should be polished with methodology and key findings.

4. Include a conceptual diagram of this study.

5. Line number 282 -“Selection of driving factors” A scientific justification is required to select driving factors.

6. Figures 2-5 – low readability; please enhance them.

7. Table 1 should be enhanced. 1Note  - what is the meaning of 1, as well as  “p<1%” and “P<5%”.

8. Discussion section should be enhanced. As well as a discussion of the specific driving factors should be included.

9. Conclusion section should be enhanced with added value.

10. Rreference style is incorrect; revise it.

https://www.mdpi.com/journal/land/instructions

Moderate editing of English language required.

Author Response

Dear reviewer:

Thank you for your decision and constructive comments on my manuscript. We have carefully considered the suggestion of Reviewer and make some changes. We have tried our best to improve and made some changes in the manuscript.

The red part that has been revised according to your comments. Revision notes, point-to-point, are given as follows,please see the attachment:

Reviewer 2 Report

The study employed geospatial analysis to identify the spatial movement within the Hangzhou metropolitan circle. The organization of the paper is commendable, with a well-detailed literature review, clear methodologies, and thorough discussion of the results. However, there are a couple of issues that need to be addressed before publication:

1) Please clarify the methodology used to generate the sub-cities. How were these sub-cities derived from the urban district? Additionally, it would be helpful to understand the distinguishing features between Xiaoshan and Central Hangzhou.

2) It would be highly beneficial if the authors included the population figures for each "sub-cities" analyzed. Given that the flow patterns may be closely linked to population numbers, this information would enhance the analysis.

The language of the paper is fine. Minor grammar and expression check is recommended.

Author Response

(The authors gave the same response as above.)

Reviewer 3 Report

The research is interesting and the paper is well structured and has some interesting findings. The aim of the paper is to investigate and analyse regional spatial structure/spatial development based on population, traffic, goods, information, and capital flows, using big data and also to provide some policy implications.

Some observations/comments:

·      Lines 50-52: you mention “China has long adopted a strategy of giving priority to the development of urban agglomerations” based on your own understanding/research, on other references or laws?

 ·      Line 55: Readers might not be familiar with the 14th Five-Year Plan of China. It would be helpful if you could mention the whole title [for example14th Five-Year Plan for (National) Economic and Social Development (2021–2025) and Long-Range Objectives through the Year 2035 (Vision 2035) of the People’s Republic of China] and a reference.

 ·      Line 109: Rearrange the order of the references.

 ·      Since one of the objectives of the article (if I have understood it correctly) is to provide policy supports for the integrated development of metropolitan areas, I think you should mention in your Introduction why integrated development is so important.

 ·      Lines 154-156: you mention that “Hangzhou metropolitan circle planning was first proposed in 2007, and then it became China’s first metropolitan circle transformation reform pilot in 2014”. It was proposed in a plan, a law (by who)?

 ·      Line 146: maybe it would be useful to write one sentence regarding the location of the Hangzhou metropolitan area (administrative divisions) (for example in which province it is located)

 ·      Line 168: In the title of Figure 1 you could add 4 “prefecture-level” cities (since in your text you also mention the 6 county-level cities). You could also add it in line 299.

 ·      Human Flow (lines 172-178): It is not exactly clear what data you are processing. Baidu migration platform displays the daily population flow and the data is collected if the users agree to share their location, but is there any information about the reson of the travel/type of travel, for work, leisure, etc.?

 ·      Lines 318-319: Figure 2 is not very easy to read

 ·      Paragraph/lines 485-510: The policy implications you propose concerns metropolitan areas in general or Hangzhou metropolitan area?

 ·      Line 504: The construction of new (and more) expressways how compatible is it with sustainable development?

 ·      You could include some more references regarding big data and spatial/urban planning.

English language is fine, no serious issues detected. Some suggestions only if the authors agree.

 ·      Lines 50-52: Maybe lines 50-52 should be in past tense since, as you state, “during the 14th Five-Year Plan period, China is striving to build a metropolitan circle system”. Unless you think that the construction of metropolitan circles is still neglected.

 ·      Lines 96-100, I think they need a rephrasing. Maybe the second sentence should start with "Moreover" or something similar. “Additionally” in line 100 it is not necessary.

 ·      Line 130: Maybe you mean “for the integrated development of metropolitan circles” instead of “integration”?

 ·      In line 146 change “four” with “4”, since you use numbers in line 149.

 ·      Lines 298-299: “… of the 24 sub-cities within the 4 (prefecture-level) cities”

·      Line 363: is described in Figure 4.

·      Lines 450-454: I think you should rephrase this sentence. In line 452 “that” is not necessary. Also you could create two sentences (the second one starting after the comma in line 451).

·      Lines 479-483: I think you should rephrase this sentence.

Author Response

(The authors gave the same response as above.)

Round 2

Reviewer 1 Report

The paper is suitable for publication.